

# Taxonomic patterns in the zoonotic potential of mammalian viruses

Alex D. Washburne[1], Daniel E. Crowley[1], Daniel J. Becker[1],
Kevin J. Olival[2], Matthew Taylor[1], Vincent J. Munster[3] and
Raina K. Plowright[1]

[1] Department of Microbiology and Immunology, Montana State University, Bozeman, MT, USA
[2] Ecohealth Alliance, New York, NY, USA
[3] National Institute of Allergy and Infectious Disease, Hamilton, MT, USA

## ABSTRACT

Predicting and simplifying which pathogens may spill over from animals to humans is a major priority in infectious disease biology. Many efforts to determine which viruses are at risk of spillover use a subset of viral traits to find trait-based associations with spillover. We adapt a new method—phylofactorization—to identify not traits but lineages of viruses at risk of spilling over. Phylofactorization is used to partition the International Committee on Taxonomy of Viruses viral taxonomy based on non-human host range of viruses and whether there exists evidence the viruses have infected humans. We identify clades on a range of taxonomic levels with high or low propensities to spillover, thereby simplifying the classification of zoonotic potential of mammalian viruses. Phylofactorization by whether a virus is zoonotic yields many disjoint clades of viruses containing few to no representatives that have spilled over to humans. Phylofactorization by non-human host breadth yields several clades with significantly higher host breadth. We connect the phylogenetic factors above with life-histories of clades, revisit trait-based analyses, and illustrate how cladistic coarse-graining of zoonotic potential can refine trait-based analyses by illuminating clade-specific determinants of spillover risk.

## INTRODUCTION

Zoonotic spillover, the transmission of pathogens from non-human vertebrate animals to humans, is a major public health challenge yet the prediction and management of which pathogens spillover is poorly understood (*Plowright et al., 2017*). Recent, high-profile spillover events, such as that of Nipah virus (*Chua et al., 2000*; *Pulliam et al., 2012*), Hendra virus (*Murray et al., 1995*; *Plowright et al., 2015*), avian influenza virus (*Fouchier et al., 2004*; *Li et al., 2014*), Ebola virus (*Baize et al., 2014*), rabies virus (*Messenger, Smith & Rupprecht, 2002*; *Schneider et al., 2009*), Chikungunya virus (*Powers & Logue, 2007*), Zika virus (*Wikan & Smith, 2016*), and West Nile virus (*Petersen & Roehrig, 2001*; *Hayes & Gubler, 2006*), among others, strongly motivate a broader understanding of the trait determinants and viral clades at risk of pathogen spillover (*Rosenberg, 2015*; *Geoghegan et al., 2016*; *Plowright et al., 2016*; *Geoghegan & Holmes, 2017*; *Lloyd-Smith et al., 2009*).

Corresponding author
Alex D. Washburne,
bigalculus@gmail.com

Emerging infectious diseases arise from a pre-existing pool of viruses with traits, life-histories, and evolutionary histories, all of which interact to determine the propensity for pathogen spillover. With an extremely large diversity of viral species in nature, understanding which clades of viruses are most prone to zoonotic spillover is critical for improving pathogen surveillance efforts and designing public health interventions across scales from reservoir hosts to humans.

A recent study examined viral traits that predicted a virus' zoonotic status; after controlling for research effort they found that predictors included the phylogenetic host breadth of the virus, genome length, and whether or not the virus was vector-borne, reproduced in the cytoplasm, or was enveloped (Olival et al., 2017). Trait-based analyses such as these are insightful and useful, but can be limited by incomplete characterization of traits associated with zoonoses and confounded or biased by non-zoonotic clades—clades with dramatically lower rates of zoonosis—whose homologous traits can all become negatively associated with zoonosis. Cladistic analyses complement trait-based analyses by identifying monophyletic clades or taxonomic groups with common patterns of zoonosis (Fig. 1).

Cladistic analyses can enhance and focus trait-based analyses to refine an understanding of the causes of viral zoonoses. Viral life-histories—the patterns and traits determining survival and reproduction—vary greatly across lineages with vastly different molecular machinery for host cell entry, viral gene expression, replication, assembly and dissemination (Jane Flint et al., 2015). General traits, such genetic material or presence of an envelope, may miss the specific molecular mechanisms, such as promiscuous receptor binding or immunoevasion, by which viral quasispecies from a particular clade can become capable of infecting humans. Furthermore, general traits found across all viruses may be less actionable for management interventions than clade-specific mechanisms. For example, knowing the particular species of mosquito, *Aedes aegypti*, responsible for transmitting many flaviviruses (Black et al., 2002) enables ecological interventions aimed at modifying *A. aegypti* population density, susceptibility to *Flavivirus* infection, and more (Morrison et al., 2008). Identifying viral clades with unusually high or low propensity for spillover can prioritize surveillance effort and generate detailed, clade-specific hypotheses about molecular mechanisms and pathogen life-history traits driving spillover risk. Furthermore, identification of non-zoonotic clades can allow refined trait-based analyses: instead of "which traits are associated with zoonosis," one can ask "which traits are associated with zoonosis for those pathogens at risk of spilling over?" Identifying molecular mechanisms and hypothesized life-history traits for clade-specific determinants of spillover can revise the list of traits used in trait-based analyses (Fig. 1).

To build off the recent study of viral traits determining spillover, we apply a new biological machine-learning method, phylofactorization (Washburne et al., 2017a, 2017b), to identify clades of mammalian viruses with high within-clade similarity in zoonotic potential and high between-clade differences. Phylofactorization is a greedy, graph-partitioning algorithm which partitions a tree, in our case the International Committee on Taxonomy of Viruses (ICTV) taxonomy tree, by identifying edges with the most
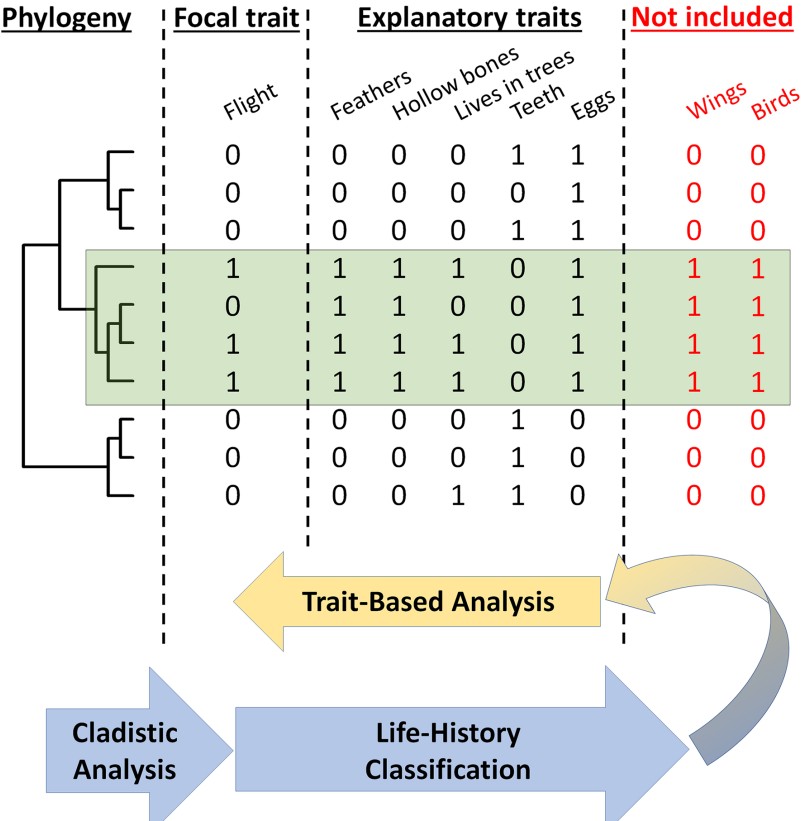

**Figure 1 Cladistic analyses complement and refine trait-based analyses of viral zoonotic potential.**
Understanding which clades of viruses spillover can improve our understanding of life-history and trait-based predictors of zoonosis. A thought experiment of predicting flight in a subset of 10 vertebrates, including reptiles, birds, and mammals reveals the importance of cladistic analyses. Trait-based analyses aim to identify explanatory traits correlated with a focal response trait, such as identifying feathers and hollow bones as associated with flight in the species observed. We propose a method outlined by the arrows. First, cladistic analysis can identify clades with a common pattern in the response variable. The clade highlighted in green—birds—has more representatives who fly than others. Then, cladistic analyses can allow life-history classification of clades to generate hypotheses about which traits (e.g., wings) and even clade labels (e.g., birds) should be included for amend trait-based analyses. Some traits, such as feathers, hollow bones, wings, and living in trees are dependent upon one-another and a consequence of life histories, and life-histories can reveal the more nuanced sets of traits, including clade-specific traits, which may underlie pathogen spillover. In this paper, for example, we find *Togaviridae* to be at risk for spilling over and discuss how previously unconsidered traits, such as variants of the E2 protein shared among *Togaviridae*, may be clade-specific determinants of spillover.

significant differences in the response variable. Using a cladistic taxonomy tree as a scaffold for identifying clades with common propensities for spillover can flexibly identify the range of taxonomic scales of interest and implicitly identify life history traits which enable viruses to spill over into humans. We analyze viruses from a recently published dataset of mammalian viruses (*Olival et al., 2017*) and replicate the paper's trait-based analysis alongside our cladistic analysis to provide a side-by-side comparison of how a cladistic analysis informs, complements and, in some cases, changes the results of trait-based analyses.

We perform phylofactorization by whether or not a clade has infected humans, discover several non-zoonotic clades, and remove the non-zoonotic clades from a replicated trait-based analysis. We then perform phylofactorization by non-human host breadth, as it is a related measurement of the host-tropism of viruses and a robust predictor of whether or not a virus can infect humans. Host-breadth phylofactorization yields a set of clades with high host breadth and high propensity to spillover. Having identified non-zoonotic clades and clades with high non-human host breadth and a high risk of zoonosis, we finish with a discussion of the life-histories of two somewhat opposite clades on the spectrum of spillover risk: the families *Togaviridae* and *Herpesviridae*. The life-history discussion is used to motivate a level of molecular and mechanistic detail which can enable clade-specific surveillance and management of pathogen spillover.

## METHODS

### Data collection

*Olival et al. (2017)* presented the most comprehensive analysis of mammalian host—virus relationships to date, spanning 2,805 mammal—virus associations (754 mammal species and 586 unique viral species from 28 viral families). We obtained these data and associated scripts for model selection from the EcoHealth Alliance HP3 GitHub repository on September 20, 2017. The ICTV taxonomy tree was constructed with arbitrary branch lengths using the R package ape version 5.4 (*Paradis, Claude & Strimmer, 2004*) in R version 3.4. Phylofactorization was implemented with the R package phylofactor version 0.0.1 available at https://github.com/reptalex/phylofactor.

### Phylofactorization

Phylofactorization partitions the phylogeny by defining a contrast function along edges (*Washburne et al., 2017a*; *Washburne, 2017*), identifying the edge which maximizes an objective of the contrast function, and iteratively partitions the tree along objective-maximizing edges to ensure non-overlapping contrasts. Phylofactorization is used to identify edges (ancestral lineages) along which major changes in zoonotic propensity or non-human host-breadth occurred, allowing us to identify clades for focused natural history analysis and inclusion as categorical variables in predictions of zoonotic risk for discrete surveillance prioritization. While phylofactorization can be used in conjunction with time-reversible ancestral state reconstruction to avoid the nested dependence of other root-to-tip ancestral state reconstruction methods, the absence of well-defined branch lengths on the ICTV taxonomy would make dubious such methods dependent upon explicit models of evolution, and so we use two-sample tests to identify edges along which major changes occurred.

For phylofactorization of Bernoulli-distributed zoonosis data, we used Fisher's exact test as a contrast function and the inverse of its $P$-value as the objective function. For phylofactorization of real-valued host-breadth data, we used a Wilcox test, and similarly used its inverse-$P$-value as the objective function. Null simulations ($n = 350$) were run for both phylofactorizations and only those factors whose objective function is larger than

the 95% quantile from null simulations of the corresponding factor were kept. This resulted in 10 factors of zoonosis, and nine factors for host breadth.

To compare the effect of non-zoonotic clades on trait determinants of spillover, we removed the first five factors from zoonosis-phylofactorization, all of which had significantly lower rates of zoonosis than the rest (*Papillomaviridae*, *Herpesviridae*, *Orthoretrovirinae*, *Nidovirales*, and *Parvovirus*). After removing non-zoonotic clades, we repeated the trait-based (*Olival et al., 2017*) classification of whether or not a virus has been observed as zoonotic using the best-fit, all data model from *Olival et al. (2017)*. Similarly, to determine whether or not the taxa from host-breadth phylofactorization predict zoonosis, we replicated the original model selection from Olival et al., but included the various taxonomic levels identified as a multilevel factor.

## Classifying zoonotic transmission of phylofactorization-derived clades

Classification of host-breadth clades with zoonotic transmission traits required curation of additional trait data for the 170 zoonoses in the data. We followed a systematic process for trait data collection (*Haddaway & Watson, 2016*). To determine human-to-human transmission, we supplemented data from *Geoghegan et al. (2016)* using eight reference textbooks in virology (https://github.com/BozemanDiseaseLab/virus_phylofactorization) and a targeted search for each zoonotic virus in GoogleScholar (see https://github.com/BozemanDiseaseLab/virus_phylofactorization for search strings). Following *Plowright et al. (2017)*, we next determined zoonotic transmission through the three primary routes of pathogen release: vector-borne transmission, reservoir excretion, and reservoir host slaughter. We expanded datasets from *Olival et al. (2017)*, *Jones et al. (2008)*, and *Plourde et al. (2017)* by screening the same texts and with a similar targeted GoogleScholar search. Pathogen release was recorded as three binary covariates for whether zoonotic transmission occurs through excretion, slaughter, or vectors; these categories are not mutually exclusive. For all zoonotic viruses within each host-breadth clade, we tabulated both the proportion of each trait (onward human transmission, vector transmission, transmission through excretion, and transmission through slaughter) and the proportion of records for which values were unknown.

## RESULTS

### Zoonotic phylofactorization and comparative trait-based analysis

Phylofactorization of whether or not a virus is zoonotic yielded 10 significant clades of various sizes and taxonomic levels (Fig. 2). The viral family *Papillomaviridae*, family *Herpesviridae*, subfamily *Orthoretrovirinae*, order *Nidovirales*, genus *Parvovirus*, and families *Caliciviridae*, *Adenoviridae*, and *Astroviridae* all had a lower fraction of zoonotic representatives compared to the paraphyletic remainder, which at the end of phylofactorization had 352 viruses of which 43.5% ($n = 153$) were zoonotic. The genera *Alphavirus* and *Deltaretrovirus* had significantly high proportions of zoonotic representatives, with 64% of the 25 *Alphavirus* species being zoonotic and 100% of the four *Deltaretrovirus* species being zoonotic.

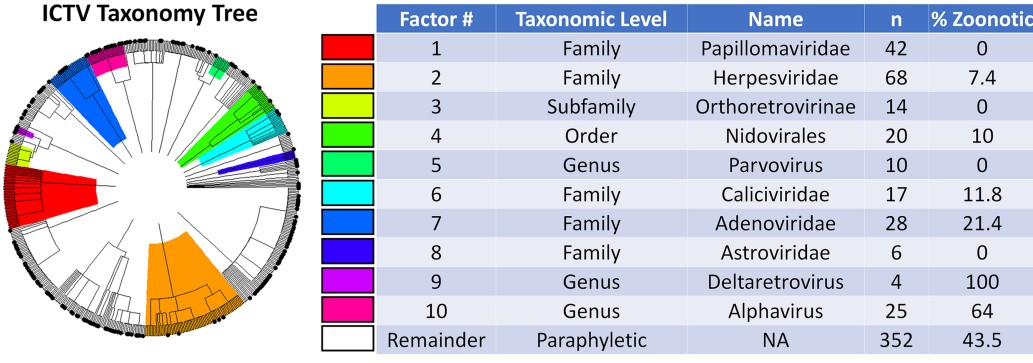

**ICTV Taxonomy Tree**

| | Factor # | Taxonomic Level | Name | n | % Zoonotic |
|---|---|---|---|---|---|
| | 1 | Family | Papillomaviridae | 42 | 0 |
| | 2 | Family | Herpesviridae | 68 | 7.4 |
| | 3 | Subfamily | Orthoretrovirinae | 14 | 0 |
| | 4 | Order | Nidovirales | 20 | 10 |
| | 5 | Genus | Parvovirus | 10 | 0 |
| | 6 | Family | Caliciviridae | 17 | 11.8 |
| | 7 | Family | Adenoviridae | 28 | 21.4 |
| | 8 | Family | Astroviridae | 6 | 0 |
| | 9 | Genus | Deltaretrovirus | 4 | 100 |
| | 10 | Genus | Alphavirus | 25 | 64 |
| | Remainder | Paraphyletic | NA | 352 | 43.5 |

**Figure 2 Phylogenetic factors of zoonosis in mammalian viruses.** Phylogenetic factorization iteratively partitions a phylogeny along edges separating species with meaningful differences. Phylofactorization of the ICTV taxonomy by a Fisher test on the fraction of zoonotic viruses identified clades with different numbers of species (n) and different rates of zoonosis (%Zoonotic). A total of 32% of the viruses in the original dataset of mammalian viruses are zoonotic, indicated as black dots on the tip of the tree. The first eight phylogenetic factors identify clades whose distinction is a significantly low rate of zoonosis relative to other viruses. Deltaretroviruses and Alphaviruses are then identified as having an unusually high fraction of zoonotic viruses. Cladistic structure in zoonotic potential can be used to identify previously ignored traits and prioritize surveillance programs.

**Table 1 Effect sizes and significance of traits pre- and post-cladistic analysis.**

| | All clades | Zoonotic clades | Non-zoonotic clades |
|---|---|---|---|
| Vector | 1.4* | 0.62 | −0.16** |
| Envelope | 0.87 | 1.2* | −0.36** |
| Cytoplasmic replication | 1.8** | 0.88 | −0.38*** |

**Notes:**
The associations between zoonotic history and traits such as enveloped, vector-borne, and cytoplasmic replication are sensitive to the inclusion/exclusion of non-zoonotic clades due to the traits' cladistic signal. Here, effect sizes (difference in linear predictors) are presented for the all viruses in the original trait-based analysis, the zoonotic clades (removal of non-zoonotic clades found in Fig. 2), and non-zoonotic clades. The direction of the effect of all three traits is conserved, but the magnitude and relative significance changes, as these traits are strongly associated with non-zoonotic clades. Asterisks indicate significance levels $*P < 0.05$, $**P < 0.01$, $***P < 0.001$ from Chi-squared tests of the deviance from generalized additive models.

Next, we excluded the top-5 taxonomic factors found to be non-zoonotic clades—*Parvoviridae*, *Herpesviridae*, *Orthoretrovirinae*, *Nidovirales*, and *Parvovirus*—and replicated the trait-based classification of zoonotic viruses from previous work (*Olival et al., 2017*). Excluding non-zoonotic clades affects the significance and explanatory power of vector-borne transmission, whether or not a virus is enveloped, and whether or not a virus replicates within the cytoplasm (Table 1). The sensitivity of results reported in previous trait-based analyses is due to strong associations between these traits and non-zoonotic clades identified through phylofactorization. Non-human host phylogenetic breadth remained a highly significant predictor of zoonosis, suggesting that among clades of viruses likely to spillover, the ability to infect disparate mammalian hosts confers a significant ability to infect humans. To perform a similar cladistic analysis of this stable predictor of a virus' ability to infect humans, non-human host breadth was used for subsequent phylofactorization to identify any phylogenetic patterns in host breadth.

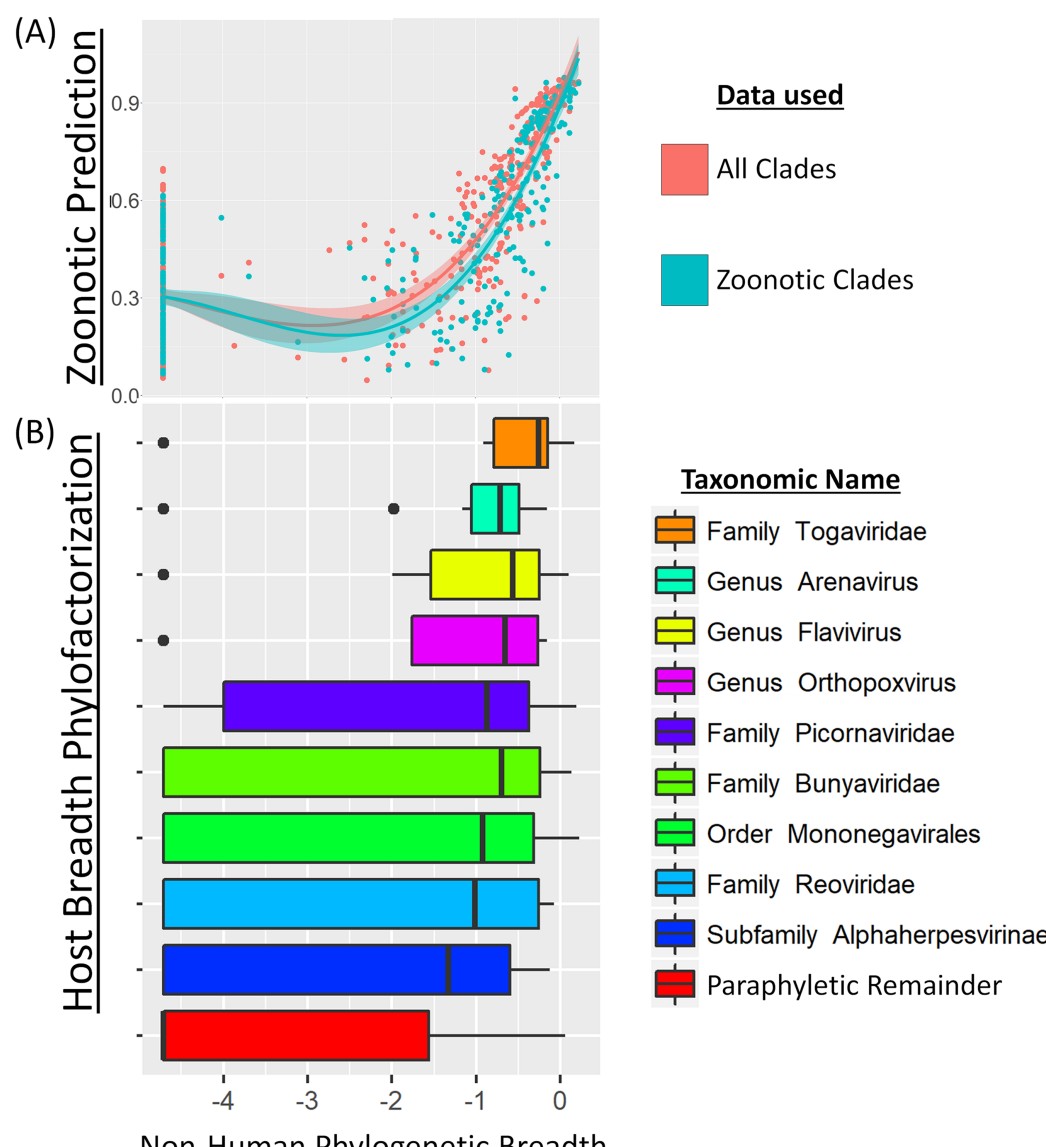

**Figure 3 Phylogenetic factors of non-human mammalian host breadth.** (A) Non-human phylogenetic distance on a cytochrome B phylogeny is a reliable predictor of zoonotic viruses, even upon removal of non-zoonotic clades identified in Fig. 2. Zoonotic prediction is the linear predictor from generalized additive modelling, plotted here against phylogenetic breadth. (B) Phylofactorization of viruses by mammalian host breadth identifies clades with high host breadth. The family Togaviridae, which contains the alphaviruses, was identified as having a significantly higher host breadth than other viruses.

## Host-breadth phylofactorization

Phylofactorization by host breadth yielded nine significant clades with a range of host breadth, each clade being partitioned as having a higher host breadth than the paraphyletic remainder. Specifically, the viral family *Togaviridae*, genus *Arenavirus*, genus *Flavivirus*, genus *Orthopoxvirus*, family *Picornaviridae*, family *Bunyaviridae*, order *Mononegavirales*, family *Reoviridae* and subfamily *Alphaherpesvirinae* all had significantly higher host breadth than the paraphyletic remainder (Fig. 3). The family *Togaviridae* contains

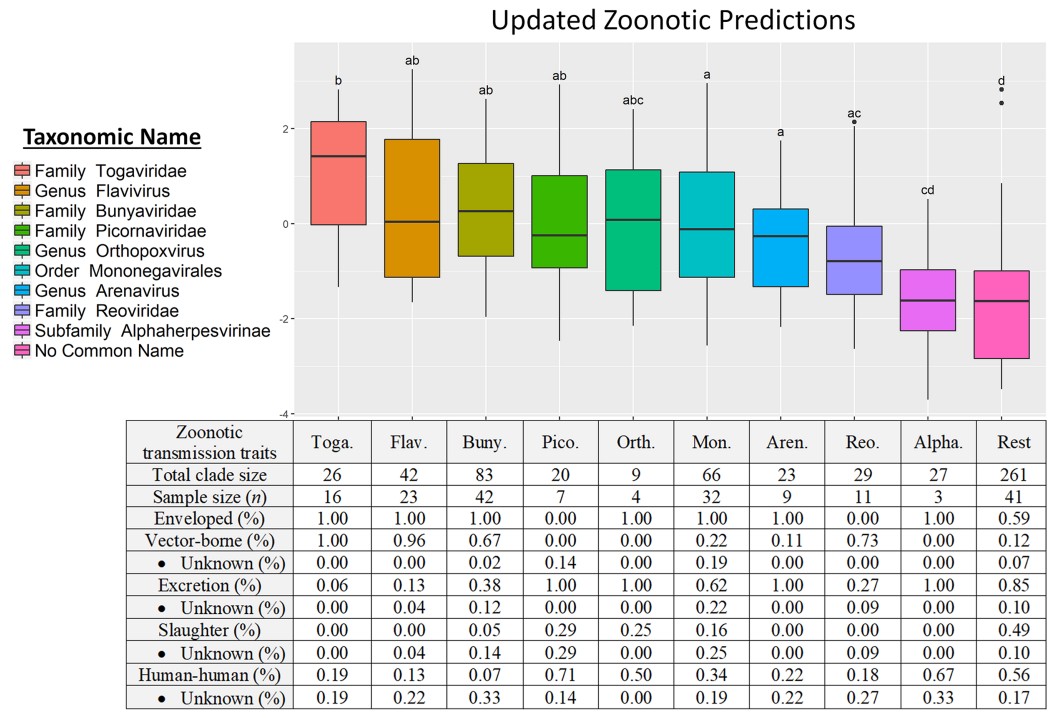

## Updated Zoonotic Predictions

**Taxonomic Name**
- Family Togaviridae
- Genus Flavivirus
- Family Bunyaviridae
- Family Picornaviridae
- Genus Orthopoxvirus
- Order Mononegavirales
- Genus Arenavirus
- Family Reoviridae
- Subfamily Alphaherpesvirinae
- No Common Name

| Zoonotic transmission traits | Toga. | Flav. | Buny. | Pico. | Orth. | Mon. | Aren. | Reo. | Alpha. | Rest |
|---|---|---|---|---|---|---|---|---|---|---|
| Total clade size | 26 | 42 | 83 | 20 | 9 | 66 | 23 | 29 | 27 | 261 |
| Sample size ($n$) | 16 | 23 | 42 | 7 | 4 | 32 | 9 | 11 | 3 | 41 |
| Enveloped (%) | 1.00 | 1.00 | 1.00 | 0.00 | 1.00 | 1.00 | 1.00 | 0.00 | 1.00 | 0.59 |
| Vector-borne (%) | 1.00 | 0.96 | 0.67 | 0.00 | 0.00 | 0.22 | 0.11 | 0.73 | 0.00 | 0.12 |
| • Unknown (%) | 0.00 | 0.00 | 0.02 | 0.14 | 0.00 | 0.19 | 0.00 | 0.00 | 0.00 | 0.07 |
| Excretion (%) | 0.06 | 0.13 | 0.38 | 1.00 | 1.00 | 0.62 | 1.00 | 0.27 | 1.00 | 0.85 |
| • Unknown (%) | 0.00 | 0.04 | 0.12 | 0.00 | 0.00 | 0.22 | 0.00 | 0.09 | 0.00 | 0.10 |
| Slaughter (%) | 0.00 | 0.00 | 0.05 | 0.29 | 0.25 | 0.16 | 0.00 | 0.00 | 0.00 | 0.49 |
| • Unknown (%) | 0.00 | 0.04 | 0.14 | 0.29 | 0.00 | 0.25 | 0.00 | 0.09 | 0.00 | 0.10 |
| Human-human (%) | 0.19 | 0.13 | 0.07 | 0.71 | 0.50 | 0.34 | 0.22 | 0.18 | 0.67 | 0.56 |
| • Unknown (%) | 0.19 | 0.22 | 0.33 | 0.14 | 0.00 | 0.19 | 0.22 | 0.27 | 0.33 | 0.17 |

**Figure 4 Phylofactorization of viruses by host range suggests a taxonomic-based surveillance prioritization scheme.** Clades identified by host-range phylofactorization vary in their fraction of zoonotic representatives, and inclusion of clades into model selection of trait-based analysis produces updated predictions for zoonosis in clades while controlling for other traits. Traits classifying the host-range clades suggest different pathways to spillover, from the vector-borne transmission of Togaviridae to environmental persistence of excreted Picornaviridae.

the genus *Alphavirus* which was also obtained in the zoonosis-phylofactorization. The subfamily *Alphaherpesvirinae*, in the family *Herpesviridae* also identified in the zoonosis-phylofactorization, was identified as having higher host breadth than the remainder of *Herpesviridae* yet the lowest host breadth of the significant, monophyletic clades obtained by phylofactorization.

In order to determine if the clades partitioned by host breadth are significant determinants of zoonotic spillover, the clades identified through host-breadth phylofactorization were then included as a multilevel factor in a replicated trait-based analysis and model selection (*Olival et al., 2017*). Incorporating the multilevel factor of host-breadth clades dramatically affected the resulting model: vector-borne transmission, viral envelopes, and cytoplasmic replication—all traits identified in model selection when not controlling for host-breadth clades—are all dropped during model selection when host-breadth clades are included as a multilevel explanatory factor.

The linear predictors of zoonotic clades under the resulting generalized additive model suggest a prioritization scheme of host-breadth clades based on the log-odds of zoonosis within-clades factored by host breadth (Fig. 4). At the top of the list is *Togaviridae* and at the bottom of the list is the paraphyletic remainder. The *Togaviridae*, *Flavivirus*, and *Bunyaviridae* clades partitioned by phylogenetic breadth are at high risk of spillover into humans; the viruses in these clades are enveloped and almost exclusively transmitted

to humans through arthropod vectors. The *Picornaviridae* and *Orthopoxvirus* had lower proportions of zoonoses but higher propensities for onward transmission in humans. *Alphaherpesvirinae* was partitioned from the paraphyletic remainder as having a slightly higher phylogenetic breadth than the remainder, which included the *Beta-* and *Gammaherpesvirinae*; however, compared to other clades partitioned by host breadth, the *Alphaherpesvirinae* have a low propensity to spillover into people.

## DISCUSSION

### Phylofactorization of mammalian viruses

Emerging infectious diseases arise from a pre-existing pool of viruses with traits, life-histories, and evolutionary histories, all of which interact to determine the propensity for spillover. Any one virus has myriad traits adapted to the virus' life cycle, ranging from interactions with a host cell that facilitate infection and replication to the processes that facilitate transmission and environmental persistence. Life-histories determine the evolutionary pressure and the constellation of traits defining a virus, and using the evolutionary tree as a scaffold for identifying clades with common propensities for spillover can implicitly identify traits and life-histories which enable viruses to spill over into humans. Modeling, prediction, and management of pathogen spillover requires a comprehensive approach that incorporates clade-specific life-histories to identify clade-specific traits that enable pathogens to jump from wildlife to people.

In this paper, we have built on a recent study of traits determining spillover in mammalian viruses (*Olival et al., 2017*) by using cladistic algorithms to partition the viral taxonomy and classify viral zoonoses. Using a modern graph-partitioning technique built for evolutionary trees (*Washburne et al., 2017a*; *Washburne, 2017*), we have partitioned the ICTV taxonomy tree and flexibly identified 10 clades of viruses across taxonomic scales with significantly extreme (high or low) propensities for zoonosis.

Removing non-zoonotic clades from the existing trait-based prediction of zoonotic viruses decreased the importance of vector-borne and cytoplasmic replication traits, but increases the importance of enveloped viruses, due to strong, negative associations of these traits with non-zoonotic clades of viruses. One trait remains a robust predictor of zoonosis: the phylogenetic breadth of non-human hosts.

Phylofactorization of the viral taxonomy by non-human host breadth yields nine clades of viruses at various taxonomic levels. These nine clades classify viruses with a range of zoonotic potential, even when controlling for host breadth. Identifying phylogenetic factors which can predict zoonosis enables researchers to take a fresh look at the problem of pathogen spillover by focusing on the life-histories of these nine clades.

Non-human host breadth is a particularly useful trait to use for phylofactorization. Zoonosis combines human:host exposure with the capacity of viruses to cross species barriers. Clades of viruses with molecular mechanisms enabling spillover into human populations, yet currently absent from humans due to infrequent exposure (i.e., viruses in remote regions or hosts, or viruses that are not shed from their hosts) will not be identified under an analysis of "zoonosis" as a response variable but can be identified by an analysis of host breadth. Likewise, using zoonoses as the outcome variable for trait-based

analyses may overemphasize the importance of traits of viruses with high human:host exposure, such as viruses that require large doses to initiate infections but manage to infect humans due to repeated exposure increasing the chance that a person is exposed to a sufficient dose to initiate infection. By performing phylofactorization on both human zoonosis and non-human host breadth, we have identified clades with risks of spillover in both humans and wildlife. There were converging lines of evidence on multiple clades in phylofactorization of both human zoonosis and host breadth: *Togaviridae* are highly zoonotic and have a broad host-range, while Herpesviridae have low rates of zoonosis and narrow host ranges; finer detail is observed in the alphaviruses' propensity for zoonosis and the *Alphaherpesvirinae* having higher phylogenetic breadth than other Herpesviridae while, compared to other clades, still having a lower propensity to spillover. We examine possible life history traits of these and other clades below.

## Life history traits of zoonotic and non-zoonotic clades

The family *Togaviridae*, containing the genus *Alphavirus*, is a family with a high host breadth and a high propensity for spilling over into humans. Togaviruses are vector-borne through mosquitoes and ticks, and such life-history may have selected generalist pathogens and produced a collection of traits shared within the lineage which enable proliferation in the blood of numerous vertebrate hosts. The Togaviruses are generalist pathogens with high host breadth and propensities for spillover, and their host breadth requires proteins which enable receptor-binding and replication in cells from different species. Identifying these determinants of host breadth in Togaviruses may focus attention on the key traits enabling pathogen spillover in this highly zoonotic clade.

It's hypothesized that Chikungunya virus, an *Alphavirus*, binds receptors ubiquitously expressed among species and cell types (*Van Duijl-Richter et al., 2015*), and that the Sindbis virus binds the laminin receptor (*Wang et al., 1992*), heparan sulfate (*Wang et al., 1992*; *Byrnes & Griffin, 1998*), and the C-type lectins DC-SIGN and L-SIGN (*Klimstra et al., 2003*), all of which are found on most cell surfaces of most animal tissues. The broad range of cellular receptors utilized by alphaviruses is attributable to a common protein shared among alphaviruses: the E2 protein (*Voss et al., 2010*). The E2 protein contains residues critical for broad host range and tissue tropism and thus may be a key trait underlying the zoonotic potential of Togaviruses. Togaviruses all contain six genes, and sequence-based surveillance targeting all Togaviruses can improve assessments of spillover risk among mammalian viruses. Furthermore, targeted sequencing of genes particular to Togavirus life history—in particular, variation in the E2 protein—may improve assessments of spillover risk within *Togaviridae* (*Klimstra et al., 2003*).

At the opposite end of the spillover spectrum from *Togaviridae* is the viral family *Herpesviridae*, which has a low propensity to be zoonotic. The subfamily *Alphaherpesvirinae* has low non-human host breadth relative to the other clades pulled out in phylofactorization, but it was identified as having a higher host breadth than the *Beta-* and *Gammaherpesvirinae* and the paraphyletic remainder of viruses from which it was partitioned.
Herpesviruses have a range of modes of transmission, often establish lifelong infections, and have several methods to evade the immune system (*Roizman, 1982*). Cross-species transmission events of Herpesviruses have been documented for closely related host species (*Woźniakowski & Samorek-Salamonowicz, 2015*), such as cases of fatal human herpesvirus infection in wild primates (*Heldstab et al., 1981*) and the converse lethal infections of humans exposed to Herpes B from macaques (*Huff & Barry, 2003*). However, herpesviruses generally show relatively strict species specificity and difficulty with cross-species transmission, including when non-human primates consume the tissues of other non-human primates (*Murthy et al., 2013*). The reasons for the species-specificity of Herpesviridae are multifactorial, but careful understanding of their life history illuminates some compelling hypotheses for clade-specific barriers to spillover.

One possibility for the species specificity of Herpesviridae is tissue tropism. *Herpesviridae* infections are often associated with terminally-differentiated cells, like neurons, and immune cells. Entry into these cells may require binding receptors which are well-known to have undergone positive selection among mammalian lineages (*Kosiol et al., 2008*). However, the generalization that *Herpesviridae* are tissue-tropic and infect immune cells is not universal; it is particularly true for *Gammaherpesvirinae*, somewhat true for *Betaherpesvirinae*, and not at all true for *Alphaherpesvirinae*. Interestingly, the *Alphaherpesvirinae* were identified from phylofactorization as having a higher host breadth than the remaining *Herpesviridae*, suggesting that tissue tropism may be a clade-specific barrier to spillover in the *Herpesviridae*.

Immune evasion also plays a major role in Herpesviridae life history and the mechanisms of immune evasion in Herpesviruses may limit their host range. For example, the ICP47 gene of herpes simplex virus type-1 is involved in immune evasion through the inhibition of TAP-mediated antigen transport; the ICP47 protein exhibits a 100-fold decrease in its binding affinity for mouse TAP compared to human TAP (*Ahn et al., 1996*). TAP-inhibition has also been documented in human cytomegalovirus (*Ahn et al., 1997*), equine herpesvirus 1 (*Ambagala, Gopinath & Srikumaran, 2004*), and bovine herpesvirus 1 (*Koppers-Lalic et al., 2003*). TAP-inhibition is only one of many mechanisms of immunoevasion within the *Herpesviridae*. Host-specificity for each member of the *Herpesviridae* might be related to mechanisms for immunoevasion, like TAP-binding, which require viral proteins interfacing with the conserved pathways yet species-specific enzyme structure underlying mammalian immune systems. The immunological barriers to spillover have been documented not just in the family *Herpesviridae*, but also in Ebola virus (*Groseth et al., 2012*) and Zika virus (*Xia et al., 2018*)—in all cases, the species-specificity of protein:protein interactions can be a barrier to spillover. For the many *Herpesviridae* whose life cycles involve latent or chronic infections, unsuccessful immunoevasion in new hosts may be especially costly.

Similar life-history case studies of viral clades partitioned here, and the clade-specific barriers to spillover, may greatly improve our understanding and surveillance of viral spillover. Clades such as the flaviviruses and bunyaviruses—both predominantly vector-borne and enveloped (Fig. 4)—corroborate the notion of generalist pathogens and each clade may have genes determining spillover which are particular to that clade.

Arenaviruses have high host breadth (Fig. 3) yet comparatively low zoonotic propensity (Fig. 4). Arenaviruses bind to α-dystroglycan protein as a receptor, enabling broad cell tropism (*Meyer, De La Torre & Southern, 2002*), yet arenaviruses contain a low proportion of viruses which have spilled over into humans. Unlike the Togaviruses, bunyaviruses, and flaviviruses, arenaviruses are transmitted to humans primarily from direct or aerosolized exposure to rodent excreta (*Emonet et al., 2007*; *Gonzalez et al., 2007*). On one hand, the limited frequency of such exposures makes rodent-human transmission inefficient and less likely to result in a human-rodent zoonosis (*Bausch & Mills, 2014*); on the other hand, the high incidence of Lassa hemorrhagic fever and onward human–human transmission of the Lassa virus suggests such transmission-related bottlenecks may be bypassed with appropriate adaptations. If exposure from reservoirs to humans is the main bottleneck for the zoonosis of many arenaviruses, the sensitivity of aerosolized arenaviruses to ultraviolet radiation, pH, and temperature may modulate the likelihood for cross-species transmission (*Stephenson, Larson & Dominik, 1984*; *Gonzalez et al., 2007*; *Sagripanti & Lytle, 2011*), and thus genetic determinants of environmental persistence may be important for spillover surveillance in arenaviruses.

The case studies and hypothesized barriers to spillover expounded here are not exhaustive nor intended to be authoritative; they are intended to remind of the complexity of viral lineages' life-histories and illustrate pipeline from cladistic analysis of spillover, life history classification of identified clades, and consideration of new traits and molecular determinants clade-specific risks of zoonosis (Fig. 1). More broadly, bottlenecks to pathogen spillover may occur at one or several of the stages in the viral life cycle and the contact process between reservoir hosts and humans. Entry, viral gene expression, replication, assembly, dissemination, and human-reservoir contact are all processes which may produce clade-specific bottlenecks to spillover. Additionally, there may be multiple stages of the viral life cycle for which viruses contain adaptations which allow them to specialize or generalize across hosts.

## CONCLUSIONS

Partitioning viruses into monophyletic clades with common propensities for spillover can simplify the problem of pathogen spillover, assist primer design for genomic surveillance (*Gardy & Loman, 2017*) and increase our resolution for understanding clade-specific barriers to spillover. We have reduced a dataset of 586 viruses into nine clades with different patterns of spillover and, likely, common sequences that can be used for primer design in sequencing-based surveillance. The different clades have different life-histories, ranging from acute infections, vector-borne transmission, envelope-mediated immune evasion and promiscuous receptor-binding in *Togaviridae* to the persistent-recurrent infections, non-vector-borne transmission and species-specific immune evasion mechanisms of *Herpesviridae*.

Understanding life-histories in a handful of clades can focus surveillance efforts on clades most likely to spill-over and target particular genes believed to underlie spillover risk within the focal clade. In vitro studies of molecular mechanisms within high-risk clades, such as the Togaviruses, flaviviruses, and bunyaviruses, can determine the risk of jump-capable pathogens arising from viral variants in reservoir hosts and identify

clade-specific life-history bottlenecks to spillover. Future trait-based analyses can incorporate life-history details obtained from cladistic analyses. Similarly, future cladistic analyses can focus study on viral life-histories to provide a more complete understanding of the traits correlated with pathogen spillover.

## ACKNOWLEDGEMENTS

The views, opinions and/or findings expressed are those of the author and should not be interpreted as representing the official views or policies of the Department of Defense or the US Government. We thank members of the Plowright and Cross groups at Montana State University and the USGS for feedback.

### Funding

This research was developed with funding from the Defense Advanced Research Projects Agency (DARPA; D16AP00113), the National Science Foundation DEB-1716698, the National Institute of General Medical Sciences of the National Institutes of Health under Award Number P20GM103474 and P30GM110732, and SERDP RC-2633. The funders had no role in study design, data collection and analysis, decision to publish, or preparation of the manuscript.

### Grant Disclosures

The following grant information was disclosed by the authors:
The Defense Advanced Research Projects Agency: D16AP00113.
The National Science Foundation: DEB-1716698.
The National Institute of General Medical Sciences of the National Institutes of Health: P20GM103474 and P30GM110732.
Strategic Environmental Research and Development Program: RC-2633.

### Competing Interests

Kevin J. Olival is employed by EcoHealth Alliance New York.

### Author Contributions

- Alex D. Washburne analyzed the data, contributed reagents/materials/analysis tools, prepared figures and/or tables, authored or reviewed drafts of the paper, approved the final draft.
- Daniel E. Crowley analyzed the data, contributed reagents/materials/analysis tools, prepared figures and/or tables, authored or reviewed drafts of the paper, approved the final draft.
- Daniel J. Becker analyzed the data, contributed reagents/materials/analysis tools, prepared figures and/or tables, authored or reviewed drafts of the paper, approved the final draft.
- Kevin J. Olival contributed reagents/materials/analysis tools, authored or reviewed drafts of the paper, approved the final draft.
- Matthew Taylor authored or reviewed drafts of the paper, approved the final draft.

- Vincent J. Munster authored or reviewed drafts of the paper, approved the final draft.
- Raina K. Plowright contributed reagents/materials/analysis tools, authored or reviewed drafts of the paper, approved the final draft.

### Data Availability

GitHub: https://github.com/dncrwlye/BZDEL/tree/master/Mammalian_Virus_Phylofactorization

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
