# Peer review of "Taxonomic patterns in the zoonotic potential of mammalian viruses"

_PeerJ, doi:10.7717/peerj.5979_

## Round 0.1 · original submission · Major Revisions

As indicated in the reviews below, a number of major changes need to be made to this manuscript before it can be reconsidered for publication. In particular, providing a better, more thorough description of phylofactorization and its uses is needed, especially with regards to its use with viruses.

Reviewer 1 ·

Basic reporting

In the presented manuscript, the authors reanalysed the virus dataset and revisited the conclusions of Olival et al., 2017 about the potential of non-human mammalian viruses to infect humans (zoonotic spillover). They used taxa of virus taxonomy as surrogate replacements of monophyletic clusters of the virus tree, which is not as readily available, to produce a phylogenetic perspective on traits associated with the dataset. The authors classified taxa as zoonotic or non-zoonotic which split the original dataset into two subsets accordingly. These subsets were then reanalysed using protocols of the Olival et al., 2017 study to assess propensity of spillover. The authors then calculated range of non-human hosts for the appropriate taxa and identified taxa with a high risk of zoonosis and a large host range as those that prone most to include viruses which could present a threat to humans. The Togaviridae, Flavivirus, and Bunyaviridae taxa were found to belong to this group, while the Herpesviridae to the opposite group with the least zoonotic potential.

This reviewer was left uncertain about differences between the conclusions of two studies, those by Olival et al., 2017 and the authors of this manuscript, respectively. Another concern is that the main conclusion may be trivial and incomplete. Just following news about emerging virus infections over the last 20 years may suffice to identify the above three virus taxa as of a high spillover potential. Also in the news are Coronavirinae, Filoviridae, and Arenaviridae, and their absence among the top runners of this analysis may warrant a comment from the authors.

The authors should also consider discussing considerable limitations of relying on virus taxonomy, instead of virus phylogeny, in this type of analysis.

To help the reader navigate the manuscript, please define zoonotic and non-zoonotic clades.

Consider replacing “breadth” with “range” in the “non-human host breadth”.

Consider replacing ICTV taxonomy with virus taxonomy, which is – by definition – ICTV approved.

All ICTV recognized taxa should be written in italic.

Discussion is too long and focused on aspects not immediately connected to the obtained results.

Experimental design

101-106: ICTV taxonomy tree should be described in detail and comparison with virus taxonomy, which is a hierarchical five-level classification. The source of this tree should be provided.

“Bernoulli-distributed zoonosis data” should be specified in the Data collection. They are never mentioned elsewhere in the manuscript.

109-119: The authors should expand on explaining the meaning of “factors” in their phylofactorization analysis. In the manuscript text, it is apparent that virus clades and taxa are treated as “factors”. Is this reading correct? If so, what is the rational to write “Phylofactorization-derived Clades” in the line 129 and elsewhere? Are they simply taxa? Other examples of factors would be useful for the reader to understand the concept application to this study.

What is the difference between partitioning “ICTV taxonomy” by using phylofactorization and listing all taxa of virus taxonomy, which are accompanied with qualifiers describing certain propensity, e.g. human, zoonotic, etc?

All taxa identified in this study are at the four ranks, from genus to order. Have species been considered?

Validity of the findings

Figure legends are not informative for understanding figures, as explained below.
Fig. 1. Current legend is nice but it does not explain this figure. What do different arrows and colors show?
Fig. 2. Legend should explain parameters of Table (Taxonomic level=rank, name-> taxon name; n-?) and details of tree (black dots, etc). How was factor number determined? Should this list be ordered by %Zoonotic? Provide a supplementary Table with the data used to produce the tree.
Fig. 3. What is “Non-human phylogenetic distance on a cytochrome B phylogeny”, and its relation to this figure? What are the units used for the X axis in panels A and B? What data were used and how were they analysed for each of these panels? What does the “Original” stand for? What does the “No Common Name” stand for? Panel labels are missing.
Fig. 4. Provide explanation for the top plot (see my comments to other figures). Provide results for the “updated” along with those for the “original” dataset. Define the dataset for the updated. What does the Total clade size stand for in the bottom Table? Are the data in this Table from the Olival et al., 2017 or this study?
Table 1. Could we say, for this Table and all study, that “Original” = “All clades” and “Phylofactor-restricted” = “Zoonotic Clades”? If not, please explain the meaning of these terms in relation to zoonotic, since you use non-zoonotic as a derived dataset.

Reviewer 2 ·

Basic reporting

The methods are not explained well (see below), but this is overall a well written manuscript with clear and engaging prose.

Experimental design

The questions that the paper addresses will be of wide interest to readers. These results would be difficult to replicate without providing additional details about the methods.

Validity of the findings

The methods used and the the authors' interpretation of their results seems sound. However, I had to do a good bit of reading outside of the material presented here to reach this conclusion.

Additional comments

In this study the authors apply a new method, phylofactorization, to investigate the properties of viruses that are prone to spillover in humans. They find that non-human host breadth is an important predictor of zoonotic potential.

The study is of an important topic, and has generated a lot of interesting hypotheses. The only real issue I have with it is that phylofactorization isn’t described very well. Three studies describing the method are cited, but only one of them has been peer reviewed. Somewhat confusingly, the one peer reviewed publication primarily concerns the analysis of microbiome data sets.

In figure 1, the authors attempt to illustrate how this method works. However, it is unclear why the traits wings and birds are excluded, given that their distribution across the tips of the tree is identical to that of feathers and hollow bones. The only trait that appears to contain any additional information about flight is “Lives in trees” in this example, and obviously many animals live in trees and don’t fly. Aside from that, wouldn’t conventional phylogenetic comparative method find a significant association between living and trees and flight? Why do we need to use phylofactorization to find a clade to restrict the analysis to? Couldn’t we have identified the clade using ancestral character reconstruction (e.g., focusing on the largest clade or clades with a high likelihood of having a common ancestor with flight)? The figure left me with more questions than answers.

By reading some additional papers I was able to get a better sense for how the method works. I found Washburne’s (Washburn et al. 2018) paper in Nature Microbiology particularly helpful. However, when presenting results using a method that very few readers will be familiar with, I believe the onus is really on the authors to 1. Explain how the method works clearly enough that most readers will understand it and 2. Explain why more widely known techniques such as PGLM analyses, concentrated changes tests of trait association, and/ or ancestral character reconstruction couldn’t have been used here.

The description of the methods is also pretty vague in general. They say that “the ICTV taxonomy tree was constructed using the R package ape.” What version of ape? What version of R? Was it a zero branch length tree, or were all branch lengths assigned a value of one? If neither of these, how were branch lengths derived? How robust is the phylofactiorization method to branch length assumptions? How were the phylofactorization analyses implemented? Presumably using the phylofactor R package. If so why isn’t the package mentioned?

Apart from these issues, I think this is an interesting study. The results suggest which taxonomic groups should be the major focus on future work to understand viral spillover, and two traits that are strong predictors of zoonotic potential within those groups. Other traits that have been shown to be important in a previous study seem primarily diagnose whether viral species are members of the taxonomic groups revealed by these analyses rather than having a function is spillover per se. These are all potentially important results. Of course, given that the “Non-zoonotic clades” contain numerous (at least 15 based on figure 2) species known to be zoonotic, understanding variation outside of the phylofactor restricted clades is still also of interest.

Cited:

Washburne, A. D., Morton, J. T., Sanders, J., McDonald, D., Zhu, Q., Oliverio, A. M., & Knight, R. (2018). Methods for phylogenetic analysis of microbiome data. Nature Microbiology, 3(6), 652.

Reviewer 3 ·

Basic reporting

This paper tackles the difficulties in predicting viral spill over. It presents a new approach, i.e. phylofactorization, for a better understanding of which viruses are more likely to jump between hosts. This method consists of re-organizing the virus taxonomy using the host range of non-human viruses and whether such viruses have infected humans. This is different to previous methods used, where virus traits were used to predict potential spill over. The authors claim that traits might be negatively associated with spill over due to the fact that the trait is shared by many different virus species and families, most of which might be non-zoonotic. Hence, the authors suggest regrouping viruses according to their spill over potential and then identify common traits within these groups.
The manuscript uses appropriate English throughout and is structured clearly, although I think the discussion is a bit too long, particularly lines 301 ff. could be cut down. Figures are specified in the correct places and of reasonable high quality. Raw data is supplied as requested. I have a few suggestions that I believe would improve the manuscript.
Figure 1. I understand this figure is used to explain the model, however, I found it difficult to associate the figure with the actual methodology used in the paper, as the figure used bird traits and the paper uses virus traits. Would it be possible to use virus traits as an example?
Table 1. I suggest using a text format instead of power point for this table. Maybe use the same format as in figure 4.

Experimental design

The experimental set up is well-described and reasonable for the question asked. Predicting viral spill over events is a difficult if not impossible task and the paper here uses a novel approach to address the issue. Methods are well described, and the data is available online for replication. The authors re-analysed an already published data set using their novel method and discuss the differences in the results.

Validity of the findings

The paper addresses the difficulties in predicting spill overs and how caution needs to be taken in consideration when analysing potential predictive traits. The authors assess these difficulties and resolve the problem partially by investigating traits common for viruses that spill over frequently in comparison to traits common for viruses that do not. The authors clearly state the different outcome in which trait is associated with spill over sing their method and previous method. However, I lack more information on which virus families are more associated with spill over using either methods. I think the authors could elaborate on that a bit more.

Additional comments

Line 13. I am missing the connection between the first and second sentence in the abstract. I suggest removing this sentence and add an introduction sentence on how current classifications might be misleading.
Line 189. ‘when’ is used twice.
Line 362. Replace ‘quasispecies’ with ‘variants.

---

## Round 0.2 · accepted · Accept

Thank-you for your quick response to questions raised by the reviewers and myself. I am happy to have this paper accepted for publication.

# Reviewer 2 ·

Basic reporting

no comment

Experimental design

no comment

Validity of the findings

no comment

Additional comments

My primary concern was that the new methods wern't described sufficiently in the paper for readers to evaluate it, and nothing peer reviewed fully describing the method seem to even be in press at the time. However, by doing a lot of research beyond what was presented I was able to satisfy myself that the methods and conclusions were sound. Now that there is an Ecological Monographs paper describing the method in press, I don't believe readers should have any issues with this. The authors have also adequately addressed my more minor concerns.

Reviewer 3 ·

Basic reporting

I am happy with the changes the authors have made and have no further comments.

Experimental design

I am happy with the changes the authors have made and have no further comments.

Validity of the findings

I am happy with the changes the authors have made and have no further comments.

Additional comments

I am happy with the changes the authors have made and have no further comments.